# The mHealth clinical decision-making tools for maternal and perinatal health care in Sub-Saharan Africa: A systematic review

Gaudensia A. Olomi[1,2,3*], Rachel Manongi[1], Charles E. Makasi[1,4], Simon Woodworth[5,6], Pendo Mlay[1,7], Karen Yeates[8], Nicola West[8], Jane E. Hirst[9,10], Michael J. Mahande[1,11], Blandina T. Mmbaga[1,3,7‡], Lottie G. Cansdale[12‡], Ali S. Khashan[5,13‡]

**1** School of Medicine, KCMC University, Moshi, Tanzania, **2** Health Department, Kilimanjaro Regional Adminstrative Secretary's Office, Moshi, Tanzania, **3** Kilimanjaro Clinical Research Institution, Moshi, Tanzania, **4** National Institute for Medical Research- Muhimbili Research Centre, Dar es Salaam, Tanzania, **5** INFANT Research Centre, University College Cork, Cork, County Cork, Ireland, **6** Cork University Business School, University College Cork, Cork, County Cork, Ireland, **7** Kilimanjaro Christian Medical Centre, Moshi, Tanzania, **8** Department of Medicine, Queen's University, Kingston, Ontario, Canada, **9** The George Institute for Global Health, Imperial College London, London, United Kingdom, **10** Nuffield Department of Women's & Reproductive Health, University of Oxford, Oxford, United Kingdom, **11** Department of Epidemiology & Biostatistics, Kilimanjaro Christian Medical University College, Moshi, Tanzania, **12** Oxford University Hospitals NHS Trust, Oxford, United Kingdom, **13** School of Public Health, University College Cork, Cork, County Cork, Ireland

☯ These authors contributed equally to this work.
‡ These authors are joint last authors on this work.
* g.olomi@kcri.ac.tz

## Abstract

### Introduction

mobile Health (mHealth) refers to use of mobile wireless technologies for health. The potential for mHealth to enhance healthcare delivery is supported by near-universal availability of mobile phones and their expanding coverage in low- and middle-income countries. This systematic review analyses the available evidence on mHealth clinical decision-making tools in maternal and perinatal health, and whether they lead to improved maternal and perinatal health outcomes in Sub-Saharan Africa (SSA).

### Methods

**Eligibility criteria:** Studies conducted in SSA describing mHealth tools piloted or used for clinical decision-making in maternal or perinatal healthcare. Exclusion criteria included mHealth tools used outside of maternal and perinatal healthcare, publications lacking sufficient detail (where information couldn't be obtained through contacting authors), articles where tools were used on a laptop or desktop computer, and articles not published in English.

**Data availability statement:** The data underlying the results of this systematic review originate from the previously published articles and publicly accessible databases. The extracted from the included articles, along with search strategies and other relevant details are within the paper and others can be found on the supportive information file.

**Funding:** The project was partly funded by an Irish Research Council (IRC), Department of Foreign Affairs COALESCE Award (COALESCE/2021/51). I received mentorship and supervision from the project team but there was no special fund allocated for this work or for publication. No additional external funding was received for this study. The funders had no role in study design, data collection and analysis, decision to publish, or preparation of the manuscript.

**Competing interests:** The authors have declared that no competing interests exist.

**Data sources:** PubMed, CINAHL, EMBASE, Global Health, and Web of Science were searched for relevant articles following a predetermined search strategy with no date restrictions. A limited grey literature search was conducted.

**Risk of bias:** We assessed the quality of included studies using the Cochrane Risk of bias 2 tool, Newcastle- Ottawa scale and COREQ. This comprehensive approach ensured a rigorous evaluation of bias and validity in our systematic review.

**Data extraction and synthesis:** Two independent reviewers screened articles and extracted data.

## Results

1119 records were screened, and 36 articles met the inclusion criteria. Fifteen mHealth tools were identified across 11 SSA countries.

## Conclusion

mHealth tools for clinical decision-making in maternal and perinatal care were found to be feasible, usable, and acceptable. They demonstrated adequate user satisfaction, and some demonstrated improvement of pregnancy outcomes. However, technologies lack scalability, with only one scaled up nationally, and few tools interacted with existing health information systems or had plans for sustainability. This review will help establish best practice for developing and scaling up mHealth clinical decision-making tools, helping to improve maternal and perinatal healthcare in SSA.

## Introduction

The United Nations' Sustainable Development Goal 3 is to 'ensure healthy lives and promote well-being for all at all ages' [1]. This includes reducing the maternal mortality ratio to less than 70 deaths per 100,000 live births and the neonatal mortality ratio to 12 per 1,000 live births by 2030 [2]. Despite maternal mortality falling by a third over the last 20 years, 287,000 women died from pregnancy-related causes in 2020, equal to one woman dying every two minutes worldwide [3,4]. Maternal mortality is a particularly acute problem in Sub-Saharan Africa (SSA), accounting for 70% of global maternal deaths [5].

mHealth (mobile health) refers to the use of mobile wireless technologies for health [6]. There are 515 million mobile phones in Sub-Saharan Africa (SSA), with 46% of the population having a mobile phone subscription in 2022 [6]. By addressing the technological weaknesses inherent in healthcare systems, mobile health technologies (mHealth) and other electronic health (eHealth) solutions have the potential to increase the quality of healthcare. However, many mHealth and eHealth initiatives have not reached their full potential as few countries have fully transitioned from paper-based medical records to electronic systems [7,8]. Global health organizations are now pursuing more comprehensive and long-term strategies to integrate e-Health and mHealth into healthcare systems in low and middle-income countries (LMICs) [8–11].

Clinical decision-making is an art and a science; clinicians use both intuition and evidence-based guidelines to make diagnostic and management decisions [12]. With the use of guidelines and protocols in medicine rising, an increasing number of clinical decision-making tools are being developed and implemented to improve health outcomes by standardizing care and improving adherence to evidence-based practice. Clinical decision-making tools can be defined as an 'electronic or non-electronic system designed to aid directly in clinical decision-making, in which characteristics of individual patients are used to generate patient-specific assessments or recommendations that are then presented to clinicians for consideration' [13]. Clinical decision-making tools can be integrated into mHealth systems, and available evidence suggests they can contribute to improved patient outcomes [14]. Digitising health systems will be one of the catalysts for achieving the SDGs related to health, and development of electronic clinical decision-making tools will be useful in raising treatment standards; however, more research into their design and implementation is critically needed [15].

mHealth is increasingly being recognized as part of effective strategies that can improve maternal and neonatal health [16,17]. Several mHealth interventions have been described in the field of maternal healthcare [18]. These have a variety of purposes, including: aiding communication between healthcare workers; collection of data; appointment reminders; and patient education [18,19]. mHealth can be utilized in clinical decision-making, improving access to evidence-based guidelines and prompting professionals to enquire and act appropriately during consultations, and such tools have been reported to improve the performance of health workers [20–22]. However, to date, there has been insufficient evidence to guide the deployment and scaling-up of mHealth interventions in LMICs, despite hundreds of pilot projects having been conducted [23]. Few mHealth pilot programs have realistic plans for expansion, and most of them are lacking interoperability or they do not involve multisectoral participation in their development and implementation [19,24]. This systematic review aims to identify and synthesise the available evidence on the usage of mHealth clinical decision-making tools by healthcare workers in maternal and perinatal health, and whether they lead to improved maternal health outcomes in SSA.

## Materials and methods

A protocol for this systematic review was registered in the PROSPERO database (CRD42023452760) and published in HRB Open Research before screening papers [25]. The review methodology has been written by the Preferred Reporting Items for Systematic Review and Meta-Analysis (PRISMA-P) guidelines [26].

### Review question

The SPIDER (Sample, Population of Interest, Design, Evaluation, Research type) was used to formulate review questions.

| SPIDER Framework | Eligibility Criteria |
| --- | --- |
| S: Sample | mHealth tools used for clinical decision-making by healthcare workers in maternal and perinatal healthcare in Sub-Saharan African countries [27] |
| PI: Phenomenon of Interest | mHealth tools used to support clinical decision-making in maternal and perinatal healthcare |
| D: Study Design | Any study, report, or publication published in English, detailing the pilot or use of an mHealth clinical decision-making tool for use in maternal and perinatal care |
| E: Evaluation | What decision was the tool designed to support?<br>Has the tool been used in clinical practice?<br>Has there been external validation of the use of the tool?<br>Is there evidence to support the use of the tool resulting in a change in quality of care, cost of care, and clinical outcomes?<br>What are the facilitators and barriers to the use of the tool? |
| R: Research Type | Qualitative and quantitative study types<br>Grey literature, including articles from scientific societies, non-governmental organizations, professional colleges, and PhD theses<br>Original articles only |

 

### Eligibility criteria

**Inclusion criteria.** To meet inclusion criteria, articles had to: (i) be conducted in Sub-Saharan Africa, as defined by the World Bank [27]; (ii) describe mHealth tools used for clinical decision-making; (iii) tools must have been used or piloted for use in maternal and perinatal healthcare, where perinatal is described up to 6 weeks postpartum; (iv) be published in English; (v) be an original article.

**Exclusion criteria.** Articles were excluded if they: described mHealth tools used outside of maternal and perinatal healthcare; if they did not include sufficient information on development and use of the tool (and if this information could not be obtained by contacting authors); for those with insufficient information, authors were contacted twice (either two emails or via email and Research Gate) to provide the needed information. Eventually if all of the efforts were not successful, this was labelled as "not described" Therefore these were not included in the analysis. Articles were also excluded if the tool was used on a laptop or desktop computer, as this falls outside the definition of mHealth; or if they were not published in English.

### Literature search

Five databases were search for relevant publications: PubMed, CINAHL, EMBASE, Global Health and Web of Science. Since "mHealth" was not a commonly used term before 2000, date constraints were not applied. A grey literature search of scientific bodies, non-governmental organisations and PhD theses was also carried out. The search strategy can be found in the appendix 1 in S1 File. It included 4 concepts related by AND function: mHealth; maternal and perinatal health-care; clinical decision making; and Sub-Saharan Africa. The search strategy was developed for one database and then adapted for use within other databases. All searches were conducted at the same time from 06/04/2023 and updated on 19/04/2024.

### Screening of articles

Citations from the databases were uploaded to EndNote and duplicates were removed. They were imported to Rayyan, a software for screening citations for systematic reviews. Articles were first screened by abstract, and those that passed abstract screening underwent full-text screening. Two reviewers (GO and LC) completed both stages of screening independently of each other, with any disputes resolved by a third author (AK). References of articles included after full-text screening were searched for further relevant studies, and these citations were uploaded to EndNote and included in citation screening.

**Data extraction.** An Excel spreadsheet was used for data extraction. We extracted data on article characteristics (title, authors, funding, journal), study type (qualitative, quantitative, number of participants, methods), mHealth features (technology, purpose, infrastructure required), and results (feasibility, usability, acceptability, and user satisfaction towards technology).

### Quality appraisal

Quality appraisal using either the Cochrane Risk of Bias 2 tool; for cluster randomised control trials we used Cochrane Collaboration's Risk of Bias Tool for Cluster Randomized Trials "appendix 2 in S2 File", We also used the Newcastle Ottawa Scale, for non-randomised trials, was attempted for studies reporting quantitative outcomes [28,29]. The Newcastle -Ottawa Scale (NOS) was designed for observational studies, originally case-control, and cohort designs based on three domains namely; selection of the study groups, comparability of the groups, and outcome/exposure assessment. The NOS has recently been adjusted for cross-sectional research designs. Several pieces of scientific data, such as Blanchard et al., (2024) [30] and Nyawo et al. (2022) highlight the distinctive features of the cross-sectional study, which have been captured by adapting NOS, whereby some of the original items like a follow-up for outcomes as it occurs in

cohort studies are not relevant to cross-sectional designs. Refer to https://www.ncbi.nlm.nih.gov/books/NBK607540/ and https://repository.up.ac.za/bitstream/handle/2263/84405/Nyawo_SystematicSuppl1_2022.pdf?sequence=2&utm_source=chatgpt.com,

As a result, we employed the Newcastle-Ottawa Scale to evaluate the quality of our observational study; however, for cross-sectional investigations, we specifically used the modified NOS.

Additionally, we used the Consolidated Criteria for Reporting Qualitative Research (COREQ) for purely qualitative studies since the NOS cannot be applied in the qualitative study design "S3 Table in S3 File".

**Data synthesis.** We used our findings as the basis for a narrative synthesis. We began by mapping studies according to the geographical location, intervention content (intended use, end-users, intervention context) and technology details (developer, required infrastructure, interoperability). We categorised articles into two tables according to whether they reported quantitative technology-impacted health outcomes "Table 1a in S4 File" or not "Table 1b in S4 File". We also tabulated technology details "Table in S5 File). We then thematically discussed outcomes (feasibility, usability, acceptability and technology-impacted pregnancy outcomes).

## Results

### Identification of eligible studies for review

The initial search was conducted on 06/04/2023 and updated on 19/04/2024. In total, 36 articles were included for review from 1,119 records screened "Fig 1" and (appendix 1 in S1 File).

### Design of studies

Table 1 characterises each paper included in the review, differentiated by whether they reported on technology-impacted health outcomes "Table in S4 File" or not "Table 1b in S4 File". Table 2 describes each technology identified. Thirty-six (36) studies describing 15 distinct interventions were identified, in use in 11 Sub-Saharan African nations (Nigeria,

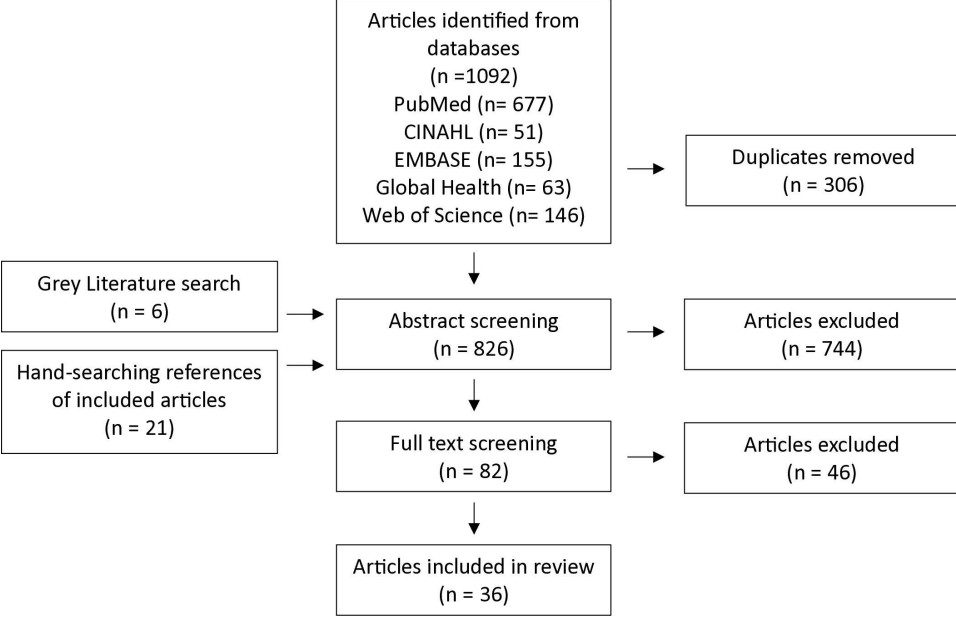

**Fig 1. Outline of the search technique.**

Mozambique, Tanzania, Madagascar, Kenya, Ethiopia, South Africa, Ghana, Burkina Faso, Malawi, and Uganda) "Tables in S4 File", "Table in S5 File".

There were 4 randomised trials, which were all cluster randomised control trials. Other studies reporting quantitative outcomes were either pre/post-intervention studies, had cross-sectional design, or mixed- methods with both qualitative and quantitative design. Qualitative studies often used focus groups and in-depth interviews to collect data, or were pilot studies reporting first use of the technology. There was one research protocol and one conference abstract for which full data could not be obtained, for Pregnancy And Newborn Diagnostic Assessment (PANDA) and mobile Partnership for Maternal, Newborn and Child Health (mPAMANECH), respectively [31].

### Intervention context

Most technologies were only reported to have been used in one country; however, CommCare, Pre-eclampsia Integrated Estimate of RiSk (PIERS) On The Move (POTM), PANDA, mobile Health for Africa (mHealth4Afrika) and electronic Parto-gram (ePartogram) all had papers describing use in more than one country, and POTM had been used outside SSA such as "*British Columbia's Women's Hospital, Vancouver, BC; Kingston General Hospital, Kingston, ON; Ottawa Hospital, Ottawa, ON; Centre Hospitalier Universitaire de Sherbrooke, Sherbrooke, QC; St. James's University Hospital, Leeds, United King-dom; Nottingham University Hospital, Nottingham, United Kingdom; King Edward Memorial Hospital, Subiaco, Western Aus-tralia; and Christchurch Women's Hospital, Christchurch, New Zealand*" [32]. mHealth for Safer Deliveries was implemented in Tanzania, with five publications [33–37], despite only this was the only technology scaled up to national level with; one paper reported that more than 320,000 pregnant women had received health visits through the programme [36].

Of the 15 technologies identified, 4 were used exclusively in the community (mHealth for Safer Deliveries, Supporting Systems to Achieve Improved Nutrition, Maternal, Newborn and Child Health (SUSTAIN), mPAMANECH, CommCare) and 8 were used only in healthcare facilities (PANDA, Bliss4Midwives (B4M), mHealth4Afrika, Nurse Assistant App (NAA), ePartogram, mHealth System, m4Change and Healthymama). Three were used in both community and facility settings (POTM, Client Data App and Clinical Decision System). A variety of end-users of technology were described, including healthcare workers (HCWs), community health workers (CHWs), midwives, traditional birth attendants (TBAs) and skilled birth attendants (SBAs). However, the formal definition of these roles or level of training they required was rarely reported, making it difficult to determine the real-world difference between the roles.

### Intervention content

All the technologies included data collection as part of their intervention. All but one intervention focussed on antenatal or postnatal visits – only the ePartogram was used exclusively during labour. Data collection items generally included demo-graphics, previous medical issues, obstetric histories, and screening to detect pregnancy complications or other medical conditions. Most of the clinical support systems aided decisions about either referral to the specialist centre or frequency of antenatal visits based on the data inputted and the presence of any danger signs "Table in S5 File". As well as referrals, POTM aided CHWs to make decisions about administering medication to treat pre-eclampsia. The ePartogram includes decision support if labour is prolonged, with the action line on the partograph prompting the clinician to take action at cer-tain time points.

POTM, PANDA and B4M also collected vital signs and could run tests to aid decision making. POTM focussed on pre-eclampsia risk management, using the PIERS algorithm developed by the same team, while B4M had tests to aid diagnosis of pre-eclampsia, gestational diabetes, and anaemia.

Outside of clinical decision making, several technologies were also able to produce appointment reminders (mHealth4Afrika, mHealth System, Client Data App), generate reports for view at regional/national level (SUSTAIN, mHealth4Afrika, mHealth System, Client Data App), and aid in patient education on obstetric health and danger signs (PANDA, mHealth System, CommCare, m4Change, Clinical Decision Support).

**Infrastructure, data storage and technical aspects.** The mERA checklist, guidelines developed by the WHO for reporting on mHealth interventions, includes information regarding infrastructure and interoperability [38]. With regards to interoperability, only mHealth for Safer Deliveries and mHealth4Afrika could communicate with the national health management information system (HMIS) using DHIS2, while the Clinical Decision Support App could connect with existing software known Open Medical Record System (OpenMRS) "Table in S5 File". Most other interventions didn't report on or discuss interoperability, or any existing electronic health systems in the intervention context.

Seven of the technologies were able to store data locally on devices and online, while 4 were only regulated to store data locally. There was no reporting on data storage for mHealth4Afrika, m4Change, Healthymama and the Clinical Decision Support system "Table in S5 File".

Regarding infrastructure, 6 technologies used Android tablets or phones, while 5 used Nokia models. Only POTM was used in the; Either iPod Touch, iPhone 3 or Android phone and was also compatible with Apple technology. NAA and mPAMANECH did not state what type of tablet or mobile phone their technology required, and mHealth4Afrika and Healthymama did not state the infrastructure the technology required. As well as a mobile phone or tablet, several of the interventions required additional devices for taking measurements of conducting additional tests. POTM required a pulse oximeter, PANDA included a blood pressure device and tests for diabetes, HIV, syphilis, malaria and anaemia, and B4M had a pulse oximeter, blood pressure device and urine analyser.

**Feasibility of the technology.** In general, the use of mHealth technologies was feasible, shown both by healthcare workers (HCWs) successfully using the technology and the ability of the different clinical decision-making systems to act. For example, POTM demonstrated the feasibility of its clinical decision-making tool by showing community health workers (CHWs) were able to recognise pre-eclampsia and give treatment as required [39] and through the PANDA app HCWs were able to successfully send 17 alerts to referral facilities [40]. Other studies showed that data entry could easily be achieved, simple tasks could be performed, and danger signs could easily be recognised [41].

However, some studies highlighted issues with feasibility, particularly regarding the use of technology. One study found lack of knowledge of mobile phone use a barrier, with HCWs unsure about how to use a touchscreen [42]. However, another study found this issue improved once staff became more familiar with the technology and had used it for longer periods [43]. Feasibility was also limited by technology infrastructure: some studies found lack of reliable electricity supply meant phones/tablets couldn't be charged, which limited technology use [44–46].

**Usability of the technology.** Usability was often assessed via qualitative means, with focus group discussions, interviews, surveys, and questionnaires, as well as through usability testing in the early stages of development. POTM used standardised usability testing (the Computer Systems Usability Questionnaire) [42] and found their technology rated highly; there were some challenges regarding the use of the mobile phone and familiarity with technology, which was echoed in other studies [47], but these could be overcome with continued use. mHealth4Afrika is an example of a technology that went through usability testing: its alpha and beta versions [48] have been reported, with the beta version having an updated user interface to view patient reports and appointments.

Important for the uptake of a new technology is that its usability is greater than the current system. Much maternal healthcare in SSA is done using paper-based records, and several studies found mHealth had increased usability compared to these, as information could more easily be retrieved, it was easy to transport records and records could more easily be shared between healthcare providers [46,49,50].

Usability was sometimes limited by the workload of those using the technology. The mHealth System in Ethiopia found reports were submitted on average 39 days after the patient encounter due to the substantial workload [51], and those using the ePartogram found the amount of data required limited its usability if HCWs had 4 or more patients [49].

**Acceptability and user satisfaction.** Most studies reported that mHealth was acceptable, both to end-users and to women clients, and studies reporting on user satisfaction found an overall positive response to technology implementation. Many healthcare workers found technology saved time on tasks; it reduced the time to conduct

healthcare visits and complete referrals [43,46]. Some technologies also had automated monthly reports, which not only reduced the reporting burden for healthcare workers but also made technology use acceptable to district health workers and directors, as this ensured reports were completed promptly [44]. However, those using B4M found the amount of data collection required by the tool increased the length of visits by 10–30 minutes [52], and poor internet connectivity caused delays in uploading data and incomplete data tasks for those using the Client Data App [44]. Some of the time delay can be due to healthcare workers having to double report in studies, inputting data both with standard-care and intervention methods, which would be remedied upon adoption of the intervention, and several others commented that increasing use and familiarity with the technology would also help the issue.

Technology also improved the perception of healthcare workers in the community, both self- and perception by other community members. mHealth for Safer Deliveries reported CHWs being more confident in when to refer as a result of the clinical decision support system [37], and another study found that women were more likely to perceive the health information as trustworthy as the application was viewed as an authoritative source [46].

Women receiving care were reported to be accepting of the different technologies across all studies where acceptability was reported. They liked the improved education given by some of the mHealth tools [50,53], the ease of communication with healthcare workers, and they felt the quality of care improved. Women exposed to PANDA reported receiving more information regarding women's health and screening and were overall more satisfied with the antenatal visits done using the tool compared to without [53]. One evaluation of SUSTAIN specifically found that women had favourable views of the clinical decision-making tool, saying 'it's a tool for guidance' and that the tool is like 'memory' to aid CHWs in conducting visits [46]. However, barriers were also noted in the included articles. Compared with traditional antenatal cards, one study found only half of the women would choose the digital health tool, stating that the ANC cards are useful to read at home and allow access to other service points [45]. Misinformation about the use of data collected has also been identified as a barrier to mHealth use [54].

**Technology-impacted pregnancy outcomes.** Eleven of the included articles collected quantitative data about the impact of the mHealth system on pregnancy outcomes "Table in S4 File". The outcome measured differed depending on the aim of the intervention – for example, SUSTAIN aimed to increase facility-based delivery, while POTM is designed to increase diagnosis of pre-eclampsia and treatment rates, so evaluated maternal and foetal morbidity and mortality [46,55]. Where reported, mHealth did improve facility delivery rates: this was reported for SUSTAIN [50], mHealth for Safer Deliveries [37], and the mHealth System in Ethiopia [56]. The latter two also found an increase in postnatal care attendance. Quality of antenatal care, calculated via client reports scores in different attributes, was increased by m4Change, who also found women were more likely to have pregnancy danger signs explained to them and have their blood pressure measured, which are important steps in recognition of pre-eclampsia [57]. However, a study on the ePartogram found no significant difference in maternal outcomes with the ePartogram compared to paper partograph but did find ePartogram use was associated with a 56% lower chance of poor foetal outcomes [49].

## Discussion

Thirty-six articles describing 15 different mHealth clinical decision-making tools were identified by this review, in use across 11 Sub-Saharan African nations. Generally, studies indicated that the mHealth tools for clinical decision making in maternal care were feasible, usable, and acceptable. They also demonstrated adequate user satisfaction, and some demonstrated a positive impact on pregnancy outcomes. However, as previous reviews have identified, there are gaps in reporting and in considering sustainability and scalability of interventions that limit conclusions around the utility of mHealth tools in maternal and perinatal care [58,59].

To the best of our knowledge, this is one of few reviews that specifically examined the use of mHealth clinical-decision-making tools in maternal healthcare in SSA. Clinical decision-making tools are becoming increasingly popular,

and using mHealth to implement them has the potential to widen their use and analyse their effects at scale. A previous review on mHealth clinical decision-making tools in SSA found that clinical decision-making tools improved client-provider relationships; clients had increased trust in the skills and knowledge of the healthcare provider, which in turn increased the confidence of the provider [60]. However, they also highlighted an important limitation of clinical decision-making tools, which is having the ability to act on the recommendation made; an importance consideration for implementation of such tools in low resource settings like SSA.

Our review is consistent with findings from other reviews on mHealth technologies [58,59], our work bring attention to these issues within the specific context of SSA, a region that has been underrepresented in the literature. One recent review of mHealth interventions in maternal, newborn and child health in LMICs similarly found a high risk of bias due to inappropriate randomisation and incomplete data. Our review reinforces these concerns, showing that many of the included articles do not report enough data to adequately assess the risk of bias in methodologies [61]. These common findings highlight the need for improved study design, reporting and transparency in mHealth interventions to ensure their effectiveness in maternal healthcare. High risk of bias also means data on whether mHealth interventions improve outcomes is uncertain, making it difficult to justify implementation on a large scale [62]. This lack of scalability results in multiple interventions being targeted at the same problem, without consideration of scalability or having learnt the lessons of previous initiative. One recent article, analysing the digital health landscape in SSA, found 983 digital tools in use, including many overlapping solutions lacking scalability and interoperability. The authors aptly called this 'e-Chaos'[63].

## Strength and limitations

**Limitations of the evidence.** While some well-designed trials were identified by our review, the majority of articles included used qualitative methods to assess technologies, without robust evaluation of their impact on maternal outcomes. Only one, mHealth for Safer Deliveries, had been scaled up to a national level, and few had plans for either scalability or sustainability. Several infrastructure issues were highlighted – lack of reliable electricity and internet connection required to consistently use an mHealth system – but few solutions were offered to this or attempts to overcome them.

Gaps in reporting leave guesswork about both the intervention context and the study design. The WHO developed the mERA checklist [38] to guide reporting of mHealth interventions, but only one included article used the checklist [44]. This makes it not only difficult to compare interventions, due to the lack of like-for-like reporting, but as the checklist includes variables such as infrastructure, platform, cost assessment, and contextual adaptability, the lack of reporting contributes to the e-Chaos mentioned above due to lack of consideration of scalability and sustainability.

**Limitations of included studies.** Many articles included did not report all the items included in our data collection – there was limited reporting on whether the technology was open source or any interoperability with existing health information systems. Authors were contacted to provide missing information, but this was often unsuccessful.

As previously reported [22], there is also a lack of high-quality evidence on the use of these technologies. Out of 36 articles included, only 4 were randomised trials, and only 11 out of 36 reported on technology-impacted pregnancy outcomes, so there remains limited evidence that mHealth clinical decision-making tools can have a positive impact on pregnancy outcomes.

**Limitations of this review.** The inclusion of this systematic review was limited to original articles and grey literature that were published in English. Quality appraisal was not possible to one study [64], an abstract, due to insufficient methodological information.

**Strengths of this review.** We used a comprehensive search strategy, with no date restrictions and conducted in several databases. We also performed a grey literature search, allowing us to capture technologies not reported in per review literature.

## Conclusion

In conclusion, this review highlights the potential of mHealth clinical decision-making tools for use in SSA. Overall, technologies were usable, feasible and acceptable to both healthcare workers and the patient population. However, for most there is limited evidence on intervention delivery at scale or plans for sustainability. Consistent, standardised reporting of interventions is critical. Shared knowledge will reduce duplication and the number of overlapping solutions, give policy makers comprehensive information about interventions and allow for scaling up and long-term sustainability of mHealth systems.

## Supporting information

**S1 File.** This is the S1 file Appendix 1: Search strategies.
(PDF)

**S2 File. This is the S2 file Appendix 2.**
(PDF)

**S3 File. This is the S3 file Table 3-Risk of bias table.**
(PDF)

**S4 File. This is the S4 Table 1; Articles describing both quantitative & Qualitative impact.**
(PDF)

**S5 File. This is the S5 Table 2; Technical aspect and intervention content.**
(PDF)

**S6 File. This is the S6 All 826 screened articles.**
(XLSX)

**S7 File. This is the S7 Data extraction.**
(XLSX)

**S8 File. This is the S8 PRISMA_2020_checklist.**
(PDF)

## Author contributions

**Conceptualization:** Gaudensia Alois Olomi, Blandina T. Mmbaga, Lottie G. Cansdale, Ali S. Khashan.

**Data curation:** Gaudensia Alois Olomi, Lottie G. Cansdale.

**Formal analysis:** Gaudensia Alois Olomi, Lottie G. Cansdale.

**Methodology:** Gaudensia Alois Olomi, Nicola West, Blandina T. Mmbaga, Lottie G. Cansdale, Ali S. Khashan.

**Resources:** Gaudensia Alois Olomi.

**Supervision:** Rachel Manongi, Charles E. Makasi, Simon Woodworth, Pendo Mlay, Karen Yeates, Jane E. Hirst, Michael J. Mahande, Blandina T. Mmbaga, Ali S. Khashan.

**Validation:** Gaudensia Alois Olomi, Lottie G. Cansdale, Ali S. Khashan.

**Visualization:** Gaudensia Alois Olomi, Charles E. Makasi, Karen Yeates, Blandina T. Mmbaga, Lottie G. Cansdale, Ali S. Khashan.

**Writing – original draft:** Gaudensia Alois Olomi, Rachel Manongi, Simon Woodworth, Pendo Mlay, Karen Yeates, Jane E. Hirst, Lottie G. Cansdale, Ali S. Khashan.

**Writing – review & editing:** Gaudensia Alois Olomi, Rachel Manongi, Charles E. Makasi, Simon Woodworth, Pendo Mlay, Karen Yeates, Nicola West, Jane E. Hirst, Michael J. Mahande, Blandina T. Mmbaga, Lottie G. Cansdale, Ali S. Khashan.

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
