## [Decision Letter · Decision Letter 0]

20 Nov 2024

PONE-D-24-41152The mHealth clinical decision-making tools for maternal and perinatal health care in Sub-Saharan Africa: A systematic reviewPLOS ONE

Dear Dr. Olomi,

Thank you for submitting your manuscript to PLOS ONE. After careful consideration, we feel that it has merit but does not fully meet PLOS ONE’s publication criteria as it currently stands. Therefore, we invite you to submit a revised version of the manuscript that addresses the points raised during the review process.

This manuscript is well written but still needs a few corrections. 

We look forward to receiving your revised manuscript.

Kind regards,

Muhammad Farooq Umer, PhD Epidemiology and Health Statistics

Academic Editor

PLOS ONE

Journal Requirements: When submitting your revision, we need you to address these additional requirements. 1. Please ensure that your manuscript meets PLOS ONE's style requirements, including those for file naming. The PLOS ONE style templates can be found at https://journals.plos.org/plosone/s/file?id=wjVg/PLOSOne_formatting_sample_main_body.pdf and https://journals.plos.org/plosone/s/file?id=ba62/PLOSOne_formatting_sample_title_authors_affiliations.pdf 2. As required by our policy on Data Availability, please ensure your manuscript or supplementary information includes the following:  A numbered table of all studies identified in the literature search, including those that were excluded from the analyses.   For every excluded study, the table should list the reason(s) for exclusion.   If any of the included studies are unpublished, include a link (URL) to the primary source or detailed information about how the content can be accessed.  A table of all data extracted from the primary research sources for the systematic review and/or meta-analysis. The table must include the following information for each study:  Name of data extractors and date of data extraction  Confirmation that the study was eligible to be included in the review.   All data extracted from each study for the reported systematic review and/or meta-analysis that would be needed to replicate your analyses.  If data or supporting information were obtained from another source (e.g. correspondence with the author of the original research article), please provide the source of data and dates on which the data/information were obtained by your research group.  If applicable for your analysis, a table showing the completed risk of bias and quality/certainty assessments for each study or outcome.  Please ensure this is provided for each domain or parameter assessed. For example, if you used the Cochrane risk-of-bias tool for randomized trials, provide answers to each of the signalling questions for each study. If you used GRADE to assess certainty of evidence, provide judgements about each of the quality of evidence factor. This should be provided for each outcome.   An explanation of how missing data were handled.  This information can be included in the main text, supplementary information, or relevant data repository. Please note that providing these underlying data is a requirement for publication in this journal, and if these data are not provided your manuscript might be rejected. 3. We note that the grant information you provided in the ‘Funding Information’ and ‘Financial Disclosure’ sections do not match.  When you resubmit, please ensure that you provide the correct grant numbers for the awards you received for your study in the ‘Funding Information’ section. 4. Thank you for stating the following financial disclosure: "The project was partly funded by an Irish Research Council (IRC), Department of Foreign Affairs COALESCE Award (COALESCE/2021/51). I received mentorship and supervision from the project team but there was no special fund allocated for this work or for publication."  Please state what role the funders took in the study.  If the funders had no role, please state: ""The funders had no role in study design, data collection and analysis, decision to publish, or preparation of the manuscript."" If this statement is not correct you must amend it as needed. Please include this amended Role of Funder statement in your cover letter; we will change the online submission form on your behalf. 5. We note that your Data Availability Statement is currently as follows: All relevant data are within the manuscript and its Supporting Information files. Please confirm at this time whether or not your submission contains all raw data required to replicate the results of your study. Authors must share the “minimal data set” for their submission. PLOS defines the minimal data set to consist of the data required to replicate all study findings reported in the article, as well as related metadata and methods (https://journals.plos.org/plosone/s/data-availability#loc-minimal-data-set-definition). For example, authors should submit the following data: - The values behind the means, standard deviations and other measures reported;- The values used to build graphs;- The points extracted from images for analysis. Authors do not need to submit their entire data set if only a portion of the data was used in the reported study. If your submission does not contain these data, please either upload them as Supporting Information files or deposit them to a stable, public repository and provide us with the relevant URLs, DOIs, or accession numbers. For a list of recommended repositories, please see https://journals.plos.org/plosone/s/recommended-repositories. If there are ethical or legal restrictions on sharing a de-identified data set, please explain them in detail (e.g., data contain potentially sensitive information, data are owned by a third-party organization, etc.) and who has imposed them (e.g., an ethics committee). Please also provide contact information for a data access committee, ethics committee, or other institutional body to which data requests may be sent. If data are owned by a third party, please indicate how others may request data access. 6. When completing the data availability statement of the submission form, you indicated that you will make your data available on acceptance. We strongly recommend all authors decide on a data sharing plan before acceptance, as the process can be lengthy and hold up publication timelines. Please note that, though access restrictions are acceptable now, your entire data will need to be made freely accessible if your manuscript is accepted for publication. This policy applies to all data except where public deposition would breach compliance with the protocol approved by your research ethics board. If you are unable to adhere to our open data policy, please kindly revise your statement to explain your reasoning and we will seek the editor's input on an exemption. Please be assured that, once you have provided your new statement, the assessment of your exemption will not hold up the peer review process. 7. We notice that your supplementary figures are uploaded with the file type 'Other'. Please amend the file type to 'Supporting Information'. Please ensure that each Supporting Information file has a legend listed in the manuscript after the references list. 8. We notice that your supplementary tables are included in the manuscript file. Please remove them and upload them with the file type 'Supporting Information'. Please ensure that each Supporting Information file has a legend listed in the manuscript after the references list.

Reviewers' comments:

Reviewer's Responses to Questions

**Comments to the Author**

1. Is the manuscript technically sound, and do the data support the conclusions?

Reviewer #1: Yes

Reviewer #2: Yes

2. Has the statistical analysis been performed appropriately and rigorously? 

Reviewer #1: Yes

Reviewer #2: Yes

3. Have the authors made all data underlying the findings in their manuscript fully available?

Reviewer #1: Yes

Reviewer #2: Yes

4. Is the manuscript presented in an intelligible fashion and written in standard English?

Reviewer #1: Yes

Reviewer #2: Yes

5. Review Comments to the Author

Reviewer #1: Good effort by the authors.

The few numbers of studies identified and analyses is an indication of the need to have more standardized and comprehensive documentations and research around mHealth in the SSA.

In line 395, the following it was mentioned that, "However, as previous reviews have identified...." It would be better to have references of some of these previous reviews.

In line 398, "To our knowledge, this is the only review written about mHealth clinical-decision-making tools

399 used in maternal healthcare in SSA." The authors need to be more cautious in the framing of their assertion. This seemed to contradict some of the discussion in line 407 - 410. The authors should check.

Between line 442 to 449, "review used a comprehensive search strategy, including a grey literature search" was mentioned as limitation and as strength. Limitations and the strengths sections of the manuscript should be revised and made clearers.

Once again, this is a very good attempt by the authors.

Best!

Reviewer #2: It was a nice oppertunity to review this manuscrpt. I have come comments below:

1. Period is missing in line 86

2. E-Health missing in line 89.

3. Reference missing in sentence of line 91 and 92.

4. What is LMICs in page 89?

5. I would suggest deleting “which was achieved as indicated in the results section part” in line 119.

6. I suggest rewriting the Materials and Methods section since there are many titles. Registration should not be the part of Method please include elsewhere. In review question section please explain in sentences rather than presenting a table. I would asl o advise not to include bullet sentences in inclusion and exclusion criteria. Please correct to Literature Search in line 151. In line 153 “Date restrictions…….please rephrase it.

7. In lines 163 and 164 authors mentioned “independent reviewers” who are also listed in the authors of the manuscripts please clarify this.

8. The quality appraisal in line 173 is not clear. Please rephrase it.

9. I suggest the table 1A, Table 1B and Table 2 should be mentioned in Data Synthesis and listed in the appendix rather than keeping in the main body of the manuscript.

10. In line 196 the “Figure 1” please include (appendix 1)

11. Design of Studies of line197 should move in Methodology and Method section.

12. I see quality Appraisal in line 238 as well. Please include in Methodology and Methods.

13. The title intervention Context is duplicated in line 245 and 269. I am not clear why it is in the results section. I also noticed that Infrastructure, data storage and technical aspects of line 289 is part of the methodology.

14. The subtitle of Results in line 310 was in italics.

15. Subtitles of Strength and limitations are in italics.

6. PLOS authors have the option to publish the peer review history of their article (what does this mean? ). If published, this will include your full peer review and any attached files.

**Do you want your identity to be public for this peer review?** For information about this choice, including consent withdrawal, please see our Privacy Policy .

Reviewer #1: No

Reviewer #2: **Yes: ** Shreedhar Acharya, NOSM University, Sudbury, Ontario

---

## [Author Response · Author response to Decision Letter 1]

18 Jan 2025

Ms. Gaudensia A. Olomi

Kilimanjaro Christian Medical University College

Box 2240 Moshi Kilimanjaro Tanzania

g.olomi@kcri.ac.tz/olomigaudensia@yahoo.com

02 January, 2025

Muhammad Farooq Umer,

PhD Epidemiology and Health Statistics,

Academic Editor,

PLOS ONE.

Dear Muhammad

Re: Response to reviewers’ comments on manuscript submission for publication titled “The mHealth clinical decision-making tools for maternal and perinatal health care in Sub-Saharan Africa: A systematic review”

We would like to thank you for giving us the opportunity to submit a revised version of the manuscript that addresses the points raised during the review process

We value the time and effort you and the reviewers took to offer insightful comments on this manuscript. We were able to make adjustments to take into account every reviewer's comment.

Here is a point-by-point response to the reviewers’ comments and concerns.

Table1 response to reviewer comments

Comment Response

1. In line 395, the following it was mentioned that, "However, as previous reviews have identified...." It would be better to have references of some of these previous reviews.

-Response; We have updated this section to include references for reviews that have identified as such. See line 353 in the manuscript (previously line 395).

2. In line 399, "To our knowledge, this is the only review written about mHealth clinical-decision-making tools used in maternal healthcare in SSA." The authors need to be more cautious in the framing of their assertion. This seemed to contradict some of the discussion in line 407 - 410. The authors should check.

Response; Thank you for raising this point as we have rechecked and confirm there is a contradiction. The assertation has been reworded “To the best of our knowledge, this is one of few reviews that specifically examined the use of mHealth clinical-decision-making tools in maternal healthcare in SSA” in line 354-355 (previously line 399). Another rewording has made in the line 364-373 (previously line 407-410) “Our review is consistent with findings from other reviews on mHealth technologies [57, 58], our work bring attention to these issues within the specific context of SSA, a region that has been underrepresented in the literature. One recent review of mHealth interventions in maternal, newborn and child health in LMICs similarly found a high risk of bias due to inappropriate randomization and incomplete data. Our review reinforces these concerns, showing that many of the included articles do not report enough data to adequately assess the risk of bias in methodologies [60]. These common findings highlight the need for improved study design, reporting and transparency in mHealth interventions to ensure their effectiveness in maternal healthcare”.

3. Between line 442 to 449, "review used a comprehensive search strategy, including a grey literature search" was mentioned as limitation and as strength. Limitations and the strengths sections of the manuscript should be revised and made clearers.

-Response; Many thanks for this correction. We have updated the limitations and strengths section, see line 405-413 (previously line 442-449). Grey literature remained in the section of strengths of this review in the study as it was written but in the limitation of this review section it has been written “The inclusion of this systematic review was limited to original articles and grey literature that were published in English”

Reviewer #2

1. Period is missing in line 86

-Response; Many thanks for your careful observation. We have amended this section to be grammatically correct, please see “However, many mHealth and eHealth initiatives have not reached their full potential as few countries have fully transitioned from paper-based medical records to electronic systems [7, 8].” line 86-88 (previously line 86).

2. E-Health missing in line 89

-Response; We agree with the reviewer, The word “e-Health” has been added, see line 89.

3. Reference missing in sentence of line 91 and 92

-Response; Thank you for this observation. We agree with this comment; the reference has been added, line 93 (previously line 91 and 92) in the manuscript.

4. What is LMICs in page 89?

-Response; Acronym expanded “low and middle-income countries (LMICs)” in line 90 (previously line 89) of the manuscript.

5. I would suggest deleting “which was achieved as indicated in the results section part” in line 119

-Response; Thank you for your observation on this review, this has been removed, line 120-121 (previously line 119) in the manuscript.

6. I suggest rewriting the Materials and Methods section since there are many titles. Registration should not be the part of Method please include elsewhere. In review question section please explain in sentences rather than presenting a table. I would asl o advise not to include bullet sentences in inclusion and exclusion criteria. Please correct to Literature Search in line 151. In line 153 “Date restrictions……. please rephrase it.

-Response; Many thanks for this feedback. We have included the registration section in the Methods as this is where PLOS One guidelines suggest to include protocols associated with the article (https://journals.plos.org/plosone/s/submission-guidelines#loc-materials-and-methods). We have decided to present our SPIDER framework as a table as it is more clearly outlines each part of the framework of question building. We have amended the eligibility criteria to be in paragraphs rather than bullet points (line 135-145), and corrected to “Literature search” line 146 (previously line 151) in the manuscript. The date restriction sentence already paraphrased, “Since "mHealth" was not a commonly used term before 2000, date constraints were not applied” see line 153 (148-150) of the manuscript. Although this is our preference, we are happy to re-consider if the editor and/or reviewer insist on this change.

7. In lines 163 and 164 authors mentioned “independent reviewers” who are also listed in the authors of the manuscripts please clarify this.

-Response; Many thanks for this clarification. By ‘independent reviewers’, we meant that the two reviewers, who are manuscript authors, reviewed the citations independently of each other, being blinded of the others’ decision until all citations were screened. We have amended this wording to make this clearer, “Two reviewers (GO and LC) completed both stages of screening independently of each other, with any disputes resolved by a third author (AK)”. line 160-161 (previously line 163-164) of the manuscript.

8. The quality appraisal in line 173 is not clear. Please rephrase it.

-Response; We have amended the quality appraisal section, “Quality appraisal using either the Cochrane Risk of Bias 2 tool, for cluster randomized control trials, or the Newcastle Ottawa Scale, for non-randomized trials, was attempted for studies reporting quantitative outcomes[28, 29]. The appraisal was incomplete and abandoned as many articles did not report sufficient data to complete the tools, see line 170-173 (previously line 173) in the manuscript.

9. I suggest the table 1A, Table 1B and Table 2 should be mentioned in Data Synthesis and listed in the appendix rather than keeping in the main body of the manuscript.

-Response; Thank you for the valuable comment. We have included mention of the tables in the data synthesis section, “We categorized articles into two tables according to whether they reported quantitative technology-impacted health outcomes (table 1a) or not (table 1b). We also tabulated technology details (table 2)” line 179-181 of the manuscript.

10. In line 196 the “Figure 1” please include (appendix 1)

-Response; Agreed, we included, ““Figure 1” (appendix 1)”. in line 187 (previously line 196) of the manuscript.

11. Design of Studies of line197 should move in Methodology and Method section.

-Response; We have kept the study design information in the results section as this describes the design of the studies identified by our literature search. As this information can only be obtained from having conducted the search, it belongs in the results section rather than the methods.

12. I see quality Appraisal in line 238 as well. Please include in Methodology and Methods.

-Response; Thank you for this comment. We have moved the appraisal section as requested. See line 170-173 (previously line 238) in the manuscript.

13. The title intervention Context is duplicated in line 245 and 269. I am not clear why it is in the results section. I also noticed that Infrastructure, data storage and technical aspects of line 289 is part of the methodology.

-Response; Many thanks for this observation. We have included these in the results section as intervention context, see line 203 (previously line 245) and intervention content in line 226 (previously line 269) are specific items extracted from the data, and hence belong in the results section – the same is true of infrastructure, data storage and technology in line 246 (previously line 289) of the manuscript.

14. The subtitle of Results in line 310 was in italics.

-Response; Thank you for your constructive comment. We have changed these subtitles so as not to be in italics “Feasibility of the technology” in line 266 (previously line 310) of the manuscript.

15. Subtitles of Strength and limitations are in italics.

-Response; Agreed. The subtitles have been changed as per suggested, “Strength and Limitations” in line 380.

We look forward to hearing from you in due time regarding our submission and to respond to any further questions and comments you may have.

Yours sincerely

Gaudensia Olomi

---

## [Decision Letter · Decision Letter 1]

4 Feb 2025

The mHealth clinical decision-making tools for maternal and perinatal health care in Sub-Saharan Africa: A systematic review

PONE-D-24-41152R1

Dear Dr. Olomi,

We’re pleased to inform you that your manuscript has been judged scientifically suitable for publication and will be formally accepted for publication once it meets all outstanding technical requirements.

Kind regards,

Muhammad Farooq Umer, PhD Epidemiology and Health Statistics

Academic Editor

PLOS ONE

Additional Editor Comments (optional):

Reviewers' comments:

Reviewer's Responses to Questions

**Comments to the Author**

1. If the authors have adequately addressed your comments raised in a previous round of review and you feel that this manuscript is now acceptable for publication, you may indicate that here to bypass the “Comments to the Author” section, enter your conflict of interest statement in the “Confidential to Editor” section, and submit your "Accept" recommendation.

Reviewer #1: All comments have been addressed

2. Is the manuscript technically sound, and do the data support the conclusions?

Reviewer #1: Yes

3. Has the statistical analysis been performed appropriately and rigorously? 

Reviewer #1: Yes

4. Have the authors made all data underlying the findings in their manuscript fully available?

Reviewer #1: Yes

5. Is the manuscript presented in an intelligible fashion and written in standard English?

Reviewer #1: Yes

6. Review Comments to the Author

Reviewer #1: The comments are addressed. Kudos for the efforts . I encourage all the authors to continue to explore more research opportunities and contribute to the body of knowledge.

Best wishes.

7. PLOS authors have the option to publish the peer review history of their article (what does this mean? ). If published, this will include your full peer review and any attached files.

**Do you want your identity to be public for this peer review?** For information about this choice, including consent withdrawal, please see our Privacy Policy .

Reviewer #1: No

---

## [Editor Report · Acceptance letter]

PONE-D-24-41152R1

PLOS ONE

Dear Dr. Olomi,

I'm pleased to inform you that your manuscript has been deemed suitable for publication in PLOS ONE. Congratulations! Your manuscript is now being handed over to our production team.

Kind regards,

on behalf of

Dr. Muhammad Farooq Umer

Academic Editor

PLOS ONE